# Assessment of Mobile Air Cleaners to Reduce the Concentration of Infectious Aerosol Particles Indoors

**Christian J. Kähler** \*,† , **Rainer Hain and Thomas Fuchs** †

Institute of Fluid Mechanics and Aerodynamics, University of the Bundeswehr Munich,
Werner-Heisenberg-Weg 39, 85577 Neubiberg, Germany

**\*** Correspondence: christian.kaehler@unibw.de

† These authors contributed equally to this work.

**Abstract:** Airborne transmission via aerosol particles without close human contact is a possible source of infection with airborne viruses such as SARS-CoV-2 or influenza. Reducing this indirect infection risk, which is mostly present indoors, requires wearing adequate respiratory masks, the inactivation of the viruses with radiation or electric charges, filtering of the room air, or supplying ambient air by means of ventilation systems or open windows. For rooms without heating, ventilation, and air conditioning (HVAC) systems, mobile air cleaners are a possibility for filtering out aerosol particles and therefore lowering the probability of indirect infections. The main questions are as follows: (1) How effectively do mobile air cleaners filter the air in a room? (2) What are the parameters that influence this efficiency? (3) Are there room situations that completely prevent the air cleaner from filtering the air? (4) Does the air cleaner flow make the stay in the room uncomfortable? To answer these questions, particle imaging methods were employed. Particle image velocimetry (PIV) was used to determine the flow field in the proximity of the air cleaner inlet and outlet to assess regions of unpleasant air movements. The filtering efficiency was quantified by means of particle image counting as a measure for the particle concentration at multiple locations in the room simultaneously. Moreover, different room occupancies and room geometries were investigated. Our results confirm that mobile air cleaners are suitable devices for reducing the viral load indoors. Elongated room geometries, e.g., hallways, lead to a reduced filtering efficiency, which needs to be compensated by increasing the volume flow rate of the device or by deploying multiple smaller devices. As compared to an empty room, a room occupied with desks, desk separation walls, and people does not change the filtering efficiency significantly, i.e., the change was less than 10%. Finally, the flow induced by the investigated mobile air cleaner does not reach uncomfortable levels, as by defined room comfort standards under these conditions, while at the same time reaching air exchange rates above 6, a value which is recommended for potentially infectious environments.

**Keywords:** indoor air cleaner; particulate filter; mobile filter devices; air exchange rate quantification; air cleaning efficiency; indoor air quality

## 1. Introduction

Note that this manuscript incorporates findings which were previously made available as preprints [1,2].

The COVID-19 pandemic placed significant attention on the airborne transmission routes of viruses via aerosols and droplets [3–5]. Aerosols are mixtures of air and particles, where the aerosol particles have a diameter ranging from 1 nm up to several 100 μm [6]. So-called droplets have a diameter of several 100 μm and larger. Droplets show a ballistic flight behavior and remain in the air for a rather short duration, which is why they can carry viruses over short distances (few meters) only [7]. This is completely different for aerosol particles, as they can move with the convective air flow for a long time and over large distances. Indoors, the aerosol particles accumulate over time in the presence of a

source, such that the infections risk is significantly higher than it is outdoors, where the aerosol is diluted strongly.

Aerosol particles are produced when breathing, speaking, singing, coughing, or sneezing [8]. The number of emitted particles per unit time depends strongly on the activity and individual factors, such as the age and the physical work load [9]. For instance, at a very high workload, almost everybody can become a so-called super-emitter [10]. In principle, there are two airborne infection routes: direct and indirect infection. Direct infection refers to infections over short distances from person to person with aerosol particles or droplets being the carrier of the viruses. Since the viral load decreases with the distance from the infected person due to turbulent mixing processes in the case of aerosol particles and due to falling to the ground in the case of droplets, people can protect themselves very effectively against this direct infection by means of social distancing. Protective barriers are another effective measure to reducing the direct infection risk. In contrast, the indirect infection mechanism refers to the infection by viruses that move with the air flow, e.g., in a room. In this case, the infection risk depends on the concentration of the virus, the susceptibility to infection (the performance of the body's defenses), the infectiousness of the virus, and the time spent in the contaminated environment. For the indirect infection mechanism, it is not necessary to have direct contact with the source of contamination. Particle-filtering respiratory masks (N95/KN95/FFP2 or better) provide a means to protecting oneself from direct as well as indirect infection since these respiratory masks reliably separate aerosol particles and droplets when air is inhaled and exhaled [7,11,12]. However, leakage flows may reduce the filtering efficiency if these masks are worn improperly [13]. Surgical masks and mouth-and-nose covers effectively decrease the direct infection risk of others by filtering large droplets and by hindering the spread of the aerosol particles in the immediate surrounding (1.5 m radius) of a person [12]. However, mouth-and-nose covers and surgical masks cannot hold back aerosol particles due to their limited filtering capability [12] and due to leakage flows at the mask–face interface [14]. Therefore, these masks do not provide adequate self-protection. As a consequence, aerosol particles accumulate indoors over time, leading to the possibility of indirect infections [11,15].

To reduce the indirect infection risks indoors, exchanging or filtering the room air is required to limit the accumulation of aerosol particles. The aerosol particle concentration $c(t)$ is directly proportional to $S$, which is the amount of exhaled particles per second, the time $t$, and inversely proportional to the room volume, $V$, and the air exchange rate, $k$:

$$c(t) = \frac{S}{k \cdot V} \cdot t. \tag{1}$$

Thus, outdoors $V$ can be considered as infinite such that $c$ approaches 0. Indoors, $c$ reaches a stationary value which is defined by $S$, $k$, and $V$. This is due to the self-inactivation of the viruses and the air exchange. For highly infectious environments, such as surgery rooms, more than 12 air exchanges are recommended by the ASHRAE [16]. Generally, the ASHRAE recommends air exchange rates of at least 6 for health care facilities, i.e., for environments where the presence of infectious persons is likely. The present study targets premises such as classrooms, offices, stores, and restaurants, where given the number of people, the presence of an infectious person is also likely. As a consequence, an air exchange rate of 6 as it is recommended for health care facilities is considered to be reasonable for these types of premises. However, a suitable air exchange rate strongly depends on the infectiousness of the specific virus and therefore has to be adapted [17].

Exchanging the room air is usually technically realized with heating, ventilation, and air conditioning (HVAC) systems. However, a large number of buildings are not equipped with HVAC systems. Opening windows requires regular action and thus interrupts work processes. In addition, the room quickly becomes cold in winter and hot in summer, and pollen, fine dust, and noise affect the room. Finally, window ventilation depends not only on the willingness of people and the number and size of existing windows, but also on the ambient conditions, such as temperature difference between inside and outside

or the wind velocity outside [18]. The temperature difference usually decreases with an increasing ventilation interval, meaning the window ventilation becomes less efficient over time when the indoor and outdoor temperatures level out. To overcome these problems, mobile room air purifiers, sometimes called air cleaners, can be used. They offer the ability to continuously filter out the aerosol particles with the desired flow rate and filtering performance. Furthermore, they are easy to install, as no holes in the walls or complex infrastructure are required. However, their effectiveness and the assessment of proper operating conditions have only been partly examined quantitatively: compact air cleaners for smaller office rooms have been shown to reduce aerosol particle concentrations over a wide range of particle sizes [19]; for larger classrooms, it has been shown that the design of the air cleaner, specifically the location of the intake, determines the optimal positioning of the device in a room, while blowing the filtered air toward the ceiling is essential for an effective distribution of the clean air across the room [20]; unlike the intermittent open window ventilation, mobile air cleaners have the advantage in that the contaminant concentration stays on a low level and does not show intermediate peaks [21,22]. However, it should be noted that mobile air cleaners neither regulate humidity nor reduce $CO_2$ concentration. A regular supply of fresh air is therefore necessary even when room air cleaners are being used.

In this experimental study, the local filtering efficiency was determined for different room configurations with a *TAC V+* mobile air cleaner from *TROTEC GmbH, Heinsberg, Germany,* being used under the following specifications: (A) a maximum volume flow rate of 1500 m$^3$/h, (B) use of a class F7 prefilter (equivalent to ISO ePM$_{2.5}$ 65% [23]) and an H14 main filter (also known as HEPA filter; for detailed specifications see [24]), (C) filtered air blowing toward the ceiling at an angle of around 25° relative to vertical coordinate in four directions, (D) an inflow only showing to one side, (E) and specific dimensions as described in Table 1. Our study aimed to answer the following questions: (1) How effectively do mobile air cleaners filter the air in a room? (2) Do parameters, such as the occupancy and the furnishing of the room, influence this efficiency? (3) Are there room situations, such as an unfavorable positioning of the air cleaner or obstacles at the ceiling, that completely prevent the air cleaner from filtering the air? (4) Does the air cleaner flow make the stay in the room uncomfortable due its induced convective flow field?

**Table 1.** Air cleaner dimensions.

| | |
|---|---|
| Housing | Width × depth × height: 500 mm × 500 mm × 1300 mm |
| Inlet (one to the front) | Width × height: 460 mm × 440 mm |
| Outlet (one to each side) | Width × height: 455 mm × 110 mm |

## 2. Materials and Methods

### 2.1. Particle Image Velocimetry

In order to assess whether it is uncomfortable to stay in a room where a mobile air cleaner is running, particle image velocimetry (PIV) was employed to determine the flow field around the device. PIV is a well-established optical velocimetry technique, where the flow is seeded with tracer particles, which are illuminated by a light source and recorded with a camera [25]. Here, planar PIV was used, where the particles are imaged in a light sheet, yielding an instantaneous vector field containing the in-plane velocity components $u$ and $v$. To seed the flow, di(2-ethylhexyl) sebacate (DEHS) aerosol particles with an average diameter of $d \approx 0.4$ µm were used since these tracers faithfully follow air flows [25,26].

For these investigations, the PIV setup consisted of 4 *LaVision Imager sCMOS* cameras, enabling the simultaneous measurement of 4 different areas in the proximity of the air cleaner. Three cameras were equipped with 35 mm *Zeiss* objective lenses and one with a 25 mm lens. The DEHS tracer particles were generated with a *PIVpart45* seeder from *PIVTEC*. Two *Quantel Evergreen 200* double-pulse lasers were used to illuminate the particles in the 4 measurement areas; one laser was illuminating the intake area close the floor (from

$y = 0 - 0.5$ m), and the other laser illuminated the outlet area (height ranging from $y = 1.1 - 1.6$ m above the floor). For the image processing and evaluation, commercial PIV software *DaVis 10.1* from *LaVision* was used.

### 2.2. Particle Image Counting for Decay Rate Estimation

The filtering efficiency is quantified in terms of the decay rate, *k*, of the test aerosol particle concentration over time. To measure this decay, the particles are distributed homogeneously in the room before the filtering process is started. In this study, the aforementioned DEHS particles served as test aerosol particles, which is a certified practice in air filter testing [27]. In particular, the longevity and the small size of the DEHS particles are essential for the air filtering efficiency assessment, as quickly evaporating droplets or settling particles would lead to bias errors in the concentration determination, yielding an overestimation of the decay rate.

Particle imaging, which is closely related to the PIV method, was used to determine the relative particle concentration. The difference to PIV is that instead of a cross-correlation evaluation being used to determine the velocity field, the particle images in the individual images are counted. A particle image is valid if its gray-scale intensity value lies above a fixed threshold of the background subtracted recordings. The evaluation of a series of recordings provides the particle image count over time, where the temporal resolution can be adapted by the recording rate, which was chosen to be 1 s for this investigation. However, since the illuminated volume size is unknown, it is not feasible to determine the absolute particle concentration from the particle image count. Hence, the concentration is normalized with the start concentration of the evaluation time interval.

The aerosol particle concentration decay can be described by an exponential function:

$$c_2 = c_1 \cdot e^{-k \cdot t} \tag{2}$$

where $c_1$ and $c_2$ are the concentrations at times $t_1$ and $t_2$, and $t = t_2 - t_1$ is the duration between these time instances. In this study, the experimental data, i.e., the particle image count over time, was fitted using the *MATLAB* 'fit' function with the following options: 'Method', 'NonlinearLeastSquares'; 'Algorithm', 'Levenberg-Marquardt'; and 'Robust', 'LAR'. Analogous HVAC systems, *k* is the equivalent to the number of air exchanges, such that the stationary contaminant concentration can be written as follows:

$$c_s = \frac{S}{k \cdot V} \tag{3}$$

where *S* is the strength of the contamination source, denoted by a volume flow rate or a particle production rate, and *V* is the room volume. Thus, for a fixed *S* and *V*, a larger decay rate, *k*, means lower stationary contaminant concentration and therefore a reduced indirect infection risk within a certain time.

The test aerosol particle concentration was determined by counting particle images. At each measurement point, the images were recorded at 1 Hz with *LaVision Imager sCMOS* cameras, each equipped with 50 mm *Zeiss* macro lenses. *Quantel Evergreen 200* lasers illuminated the test aerosol particles.

Representing a classroom, an office, a restaurant, or other public premises, Figure 1 shows the sketch of the measurement positions in an empty rectangular 80 m$^2$ room with a height of 2.5 m (*config. I empty*). Here, the filtering efficiency was measured simultaneously at 6 different locations, all situated 1.5 m above the floor: 3 along the axis of symmetry (measurement locations mp1–mp3) and 3 in one half of the room (measurement locations mp4–mp6). To obtain an estimate of how strong the influence of the air cleaner position impacts the filtering efficiency, the air cleaner was set up at two different positions: position A can be regarded as the most favorable position, while position B is less optimal, since it is located in the corner and the outlet air jet is blocked by a ceiling light. Additional measurements were carried out in which the 80 m$^2$ room was equipped with 24 tables with

chairs and bags being in a classroom configuration (*config. I classroom*), with mp3 remaining as the sole measurement point. Furthermore, to assess the influence of the room geometry, an elongated hallway (*config. II*), as illustrated in Figure 2, was investigated. In the latter configuration, the filtering efficiency was measured at two locations (mp1 & mp2). Both rooms used for the studies had hard floors rather than carpets.

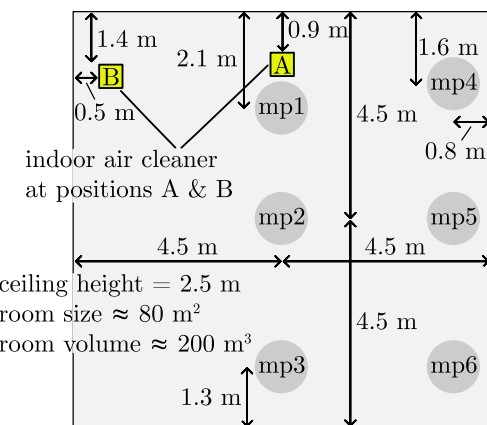

**Figure 1.** *Config. I*: the 80 m² room and the measurement locations (mp1–mp6). Figure not to scale with Figure 2.

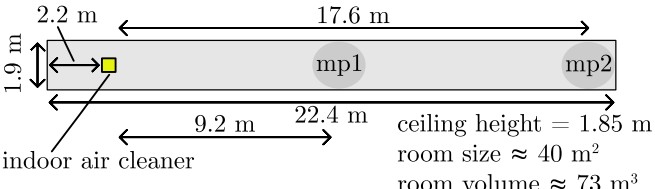

**Figure 2.** *Config. II*: the elongated hallway with the measurement locations mp1 and mp2. Figure not to scale with Figure 1.

## 3. Results

### 3.1. Comfort of Stay

The comfort of stay in a room depends on several parameters. For longer stays, these are, in particular, the temperature, the air flow velocity as well as the air humidity. In most cases, the air flow will be turbulent. The degree of turbulence in connection with the average velocity plays an important role in well-being. At higher temperatures, such as in the summer time, larger turbulence values can be assumed for good cooling comfort. Recommendations for the mentioned quantities can be taken from EN 16798-1 [28]. According to this, a mean flow velocity of 0.15 m/s with a turbulence level of 40%, is considered to be acceptable in the occupied zone at an air temperature of 22 °C. Figure 3 shows the magnitude of the average velocity (color-coded) at the center plane of the indoor air cleaner operated at a flow rate of 1500 m³/h, corresponding to the maximum flow rate of the device. The filtered air is blown upward while the air is sucked into the mobile air cleaner close to the floor. The turbulence (color-coded) is illustrated in Figure 4. It becomes clear that the values are only increased in the immediate vicinity of the air cleaner. From a distance of about 0.5 m, the values are below the threshold so that a comfortable stay is possible (except for the direct proximity of the outlet jet).

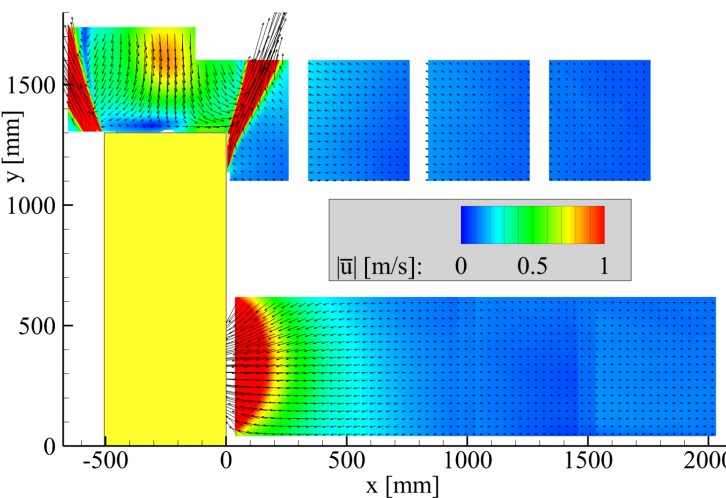

**Figure 3.** Magnitude of the mean in-plane velocity at an air cleaner flow rate of 1500 m$^3$/h. The vectors visualize the direction and strength of the air flow. The color quantitatively illustrates the magnitude of the flow velocity.

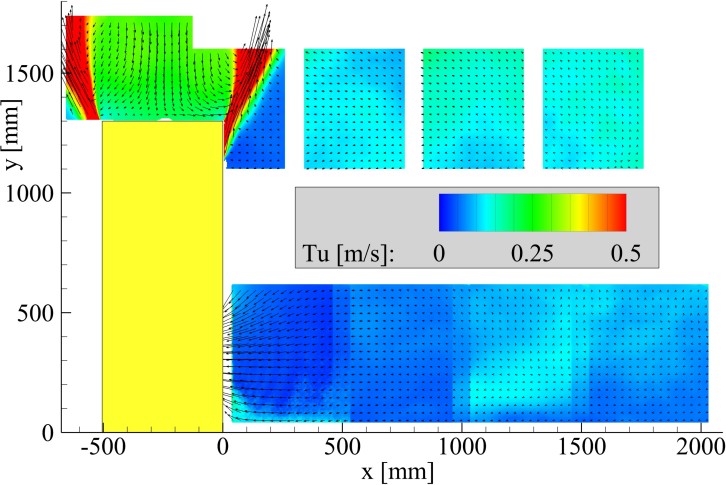

**Figure 4.** Mean turbulent velocity fluctuations at an air cleaner flow rate of 1500 m$^3$/h. The vectors visualize the direction and strength of the turbulent air flow. The color quantitatively illustrates the magnitude of the turbulence.

Apart from the air flow, people are sensitive to noise, particularly in situations where concentration is required, as it is the case in school or university and also in offices. EN 16798-1:2019 [28] provides recommendations on the maximum sound levels under different conditions. Thus, the noise emitted by the mobile air cleaner should remain at an acceptable level. An acoustically slightly optimized and commercially available version of the air cleaner was analyzed by means of a calibrated *MiniDSP UMIK-1* microphone positioned at $x = 2000$ mm and $y = 1300$ mm (see Figure 3 for the coordinate system). A sound pressure below 50 dB(A) at a flow rate of 1400 m$^3$/h was measured. This value was elevated to what is considered to be acceptable for ventilation systems in classrooms or offices [28]. Since noise is usually not an acute threat, it is important to consider whether a little more noise is acceptable, for example, to protect against a potentially deadly virus. It must also be considered that opening windows also leads to an increase in the noise level. In addition, open windows may cause unpleasant temperature changes and create drafts. Moreover, the regular opening of windows takes work, so it is often omitted. Moreover, if it is strictly necessary to stay within the recommended maximum noise limits, there is still the option to operate multiple air cleaners at flow rates of around 600 m$^3$/h while at increased purchase costs due to the number of devices.

### 3.2. Air Cleaner Filtering Efficiency

The air cleaner filtering efficiency analysis answers different research questions ranging from the general suitability of mobile indoor air cleaners for particle filtering to specific influences of room occupancy and furniture. With the empty 80 m$^2$ room serving as a baseline, (*config. I empty*), Figure 5 shows the normalized aerosol particle concentration over time for different air cleaner flow rates at measurement point mp3; see. Figure 1 for reference of the measurement point locations. The higher the flow rate is, the larger is the decay rate (colored dots), as provided in the legend of Figure 5. A reference test case without air cleaning is shown for comparison (black dots), with $k$ values on the order of 0.1 1/h, to prove that the leakage flow effects of the test room or settling of the aerosol particles over time are negligible. Running the air cleaner at 1000 m$^3$/h without any filters (gray dots) results in a slight increase of the decay rate: from $k = 0.05$ 1/h to $k = 0.15$ 1/h at mp3. This is due to particle separation by the fan as well as the deposition on surfaces. The result confirms that it is indeed the filtering process that removes the particles from the indoor air and not some other mechanism.

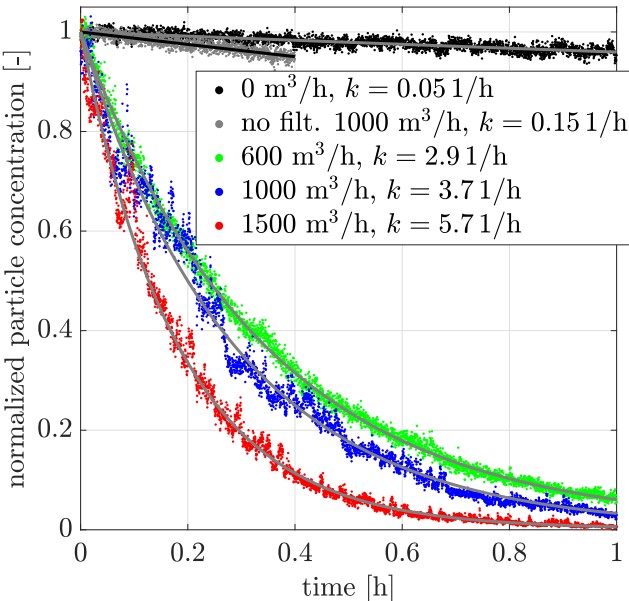

**Figure 5.** Normalized particle concentrations at mp3 (*config. I empty*) over time. Two reference cases are shown, i.e., air cleaner turned off (1, black dots) and air cleaner running at 1000 m³/h without any filters (2, gray dots). The 3 colored dots show the decay of the normalized particle concentrations over time at different volume flow rates. Increased flow rates result in increased $k$ values, ranging from $k = 2.9$ 1/h at 600 m$^3$/h up to $k = 5.7$ 1/h at 1500 m$^3$/h.

For *config. I empty*, Table 2 provides an overview of the decay rates and half-lives, i.e., the time it takes the concentration to halve in normal filtering operation for mp1–mp6. The filtering efficiency is distributed relatively homogeneously across the room, which is in accordance with a previous assessment of an air cleaner (air purifier) for smaller sized rooms [19]. Relative to mp1, situated closest to the air cleaner, the cleaning efficiency drop does not exceed 15%. Positioning the air cleaner in an unfavorable location, i.e., B instead of A as shown in Figure 1, yields an efficiency penalty of around 20% (columns 2 vs. 4 in Table 2). At location B, two of the four outlet jets of the air cleaner are facing the wall such that the filtered air cannot spread as easily along the ceiling as required for a good filtering performance. Moreover, a third outlet jet is somewhat hindered in distributing along the ceiling by the ceiling lights [1].

**Table 2.** Test aerosol particle concentration decay rates and half-lives for *config. I empty* for different flow rates.

|  | 600 m³/h Position A | 1000 m³/h Position A | 1500 m³/h Position A | 1000 m³/h Position B |
|---|---|---|---|---|
| mp1 | 3.3 ┃ 0.21 | 4.4 ┃ 0.16 | 6.2 ┃ 0.11 | 3.6 ┃ 0.19 |
| mp2 | 3.0 ┃ 0.23 | 3.9 ┃ 0.18 | 6.1 ┃ 0.11 | 3.6 ┃ 0.19 |
| mp3 | 2.9 ┃ 0.24 | 3.7 ┃ 0.19 | 5.7 ┃ 0.12 | 3.4 ┃ 0.21 |
| mp4 | 3.1 ┃ 0.21 | 4.3 ┃ 0.16 | 6.3 ┃ 0.11 | 3.4 ┃ 0.20 |
| mp5 | 2.9 ┃ 0.24 | 4.0 ┃ 0.17 | 6.2 ┃ 0.11 | 3.4 ┃ 0.21 |
| mp6 | 3.0 ┃ 0.23 | 3.9 ┃ 0.18 | 5.9 ┃ 0.12 | 3.3 ┃ 0.21 |
|  | decay rate $k$ [1/h] ┃ half-life $T_{1/2}$ [h] | | | |

Comparing the measured decay rates with the ideal values of a perfect mixing ventilation, computed as $k_{\mathrm{ideal}} = $ (air cleaner flow rate)/(room volume), reveals that $k$ does not increase linearly with the volume flow rate (note that $k$ can exceed 1; this regimen is referred to as displacement ventilation). Averaging the values of mp1–mp6 yields $k_{600}/k_{600,\mathrm{ideal}} \approx 1.01$, $k_{1000}/k_{1000,\mathrm{ideal}} \approx 0.81$ and $k_{1500}/k_{1500,\mathrm{ideal}} \approx 0.81$. Modified turbulent mixing properties at different flow rates/velocities can be considered as the cause for the relative drop in the air filtering efficiency [29]. For ventilated rooms, strong inlet jets result in a flow regime transitioning from the efficient displacement ventilation toward an inflow-dominated regime, which becomes increasingly insensitive with respect to the flow rate [30]. This is also the case for this air cleaner, where the flow rate is linearly proportional to the outlet jet speed since the outlet cross-section is constant.

Moving away from the empty room to the less generic configuration of a furnished room without people (*config. I classroom*) results in a decay rate of $k = 4.4$ 1/h at the single measurement point mp3 at a flow rate of 1000 m³/h. Compared to the empty room result at mp3, the decay is elevated, which may most likely be attributed to the fact that the filtered air supply is larger above the table level since the outlet jets of the air cleaner blow toward the ceiling. Although there is no direct evidence for this, the local $k$ value below the table top height level should be smaller, as the flow rate is constant between the two test cases.

*Config. I classroom* with an additional 13 people sitting in the room, separated by protection walls, yielded a decay rate of $k = 4.7$ 1/h at a flow rate of 1200 m³/h. Clearly, these results demonstrate that mobile air cleaners are suitable devices for filtering the air in fully furnished and occupied rooms without fundamentally altering the filtering capability relative to an empty room. It is also important to note that breathing and movement of people are basically good for the efficient reduction of aerosol particle concentration, as the additional turbulence provides improved mixing of indoor air. In addition, the increased mixing causes a reduction in the local viral load, which in turn causes a reduction in the risk of infection.

Fundamentally changing the room geometry from a squared shape toward an elongated hallway (*config. II*), where one side is around one order of magnitude shorter than the other one (see Figure 2), significantly alters the filtering efficiency. The $k$ values for the hallway are listed in Table 3 along with the natural decay rates since they are somewhat larger due to the stronger leakages in this configuration which was set up in a wind tunnel sealed at both ends. The measurement point closer to the air cleaner, mp1, shows a slightly elevated filtering efficiency, in particular if the natural decay is taken into account, which has to be subtracted from $k$ values to calculate the net decay rate. As it is the case for *config. I empty*, the relative filtering efficiency decreases with the flow rate $k_{600}/k_{600,\mathrm{ideal}} \approx 0.48$; $k_{1000}/k_{1000,\mathrm{ideal}} \approx 0.34$ and $k_{1585}/k_{1585,\mathrm{ideal}} \approx 0.32$. Evidently, the elongated shape of the room results in a significantly reduced relative filtering efficiency, which is roughly halved for *config. II* as compared the square shaped room. However, the fact that he hallway setup was not symmetrical (see Figure 2), i.e., the air cleaner was not placed in the middle of the corridor, must be taken into account. The symmetrical outlet of the air purifier results in

approximately equal volume flow rates of purified air into the short and the long part of the corridor. Thus, the volume of purified air supplied to the long part of the corridor is only about half of the nominal volume flow rate. Accordingly, it can be assumed that the air exchange rate in the short part of the corridor is significantly higher than in the long part of the corridor. However, this consideration does not include the fact that the intake is facing toward the longer hallway side, where the measurement points are situated.

**Table 3.** Test aerosol particle concentration decay rates and half-lives for *config. II* for different volume flow rates.

|  | 600 m$^3$/h | 1000 m$^3$/h | 1585 m$^3$/h | 0 m$^3$/h, reference |
|---|---|---|---|---|
| mp1 | 3.9 ∣ 0.18 | 4.7 ∣ 0.15 | 7.1 ∣ 0.10 | 0.3 ∣ 2.32 |
| mp2 | 4.0 ∣ 0.17 | 4.7 ∣ 0.15 | 6.9 ∣ 0.10 | 0.6 ∣ 1.17 |
| | | decay rate $k$ [1/h] ∣ half-life $T_{1/2}$ [h] | | |

## 4. Discussion

With reference to the research questions posed in the introduction, it can be seen that the measurement of filter efficiency in a square room showed that perfect mixed ventilation is achieved, i.e., the relative filter efficiency reaches a value of about 1 in the whole room. Toward the maximum of the air cleaner flow rate, the relative efficiency decreases to 0.8, while an elongated room shape (side wall length ratio: 10/1) yields relative filtering efficiencies below 0.5. To compensate for the efficiency penalty, the flow rate or the number of air cleaners can be increased. Most importantly, the filtering efficiency study showed that the fully furnished and occupied (13 people) room in a classroom-like configuration did not show a filtering efficiency penalty. It is rather the other way around, so that the air exchange above the tables is higher than in the more blocked area below the tables. Furthermore, the enlarged turbulence level due to the breathing and moving of the people improves the filtering efficiency. Furthermore, placing the air cleaner device in the corner of the room to not let the outlet jets flow freely along the ceiling, which can be considered the least favorable operational circumstance, introduced a filtering efficiency penalty of only 20%, which still lies in an acceptable range to ensure a sufficient filtering capability. Thus, there is not really a situation that completely prevents the air cleaner from filtering the room air.

Apart from the research questions that were answered by the filtering efficiency study, the flow field analysis demonstrated that up to the highest flow rate of the device of 1500 m$^3$/h, the sum of the average velocity magnitude and the turbulence induced by air cleaner did not reach uncomfortable levels at distances of more than 0.5 m from the device. The limiting factor for the comfort of stay is rather the noise emission, which, although it is only a qualitative statement, means that the air cleaner used in this study should only be operated to a flow rate of 1200 m$^3$/h.

From these various filtering efficiency measurements and the comfort of stay assessment, it becomes clear that it is not sufficient to only look at the nominal air cleaner flow rate to decide whether the desired $k$ value is met. Instead, the device needs to have reserves to compensate for unfavorable room geometries, unfavorable air cleaner positions, and uncomfortable noise levels; the latter effectively reduces the maximum flow rate. Moreover, the room volume needs to be taken into account as evident from Equation (3). For a large volume $V$, the stationary concentration of a contaminant remains relatively low, even if the $k$ value is low. In particular, this applies to premises with large ceiling heights, such as churches and concert halls, among others. However, this does not mean that an indirect infection is unlikely in rooms with high ceilings. Since a contaminant source generates high local concentrations of infectious aerosol particles, the surrounding of a such a source is critical, even in a very large room. In contrast, it becomes also clear from Equation (3) that small rooms are the most dangerous in terms of contaminant concentrations and therefore require a large $k$ value.

In general, this experimental study was able to demonstrate that mobile air cleaners are suitable devices for filtering the room air in premises that are not equipped with an HVAC system, providing a means to reduce the indirect infection risks by reducing the stationary contaminant concentrations. Since these devices are easily to procure and install and do not require any special infrastructure (no wall penetrations, no large electronic power, etc.) they are very well suited to quickly and easily reducing the risk of infection whenever needed. Furthermore, the devices also ensure that the exposure to fine dust and pollen is reduced. Thus, even in normal times, they can make a useful contribution to the health of the people in a room.

**Author Contributions:** Conceptualization, C.J.K., R.H. and T.F.; methodology, C.J.K., R.H. and T.F.; software, T.F.; validation, T.F.; investigation, T.F. and R.H.; data curation, R.H. and T.F.; writing—original draft preparation, C.J.K., R.H. and T.F.; writing—review and editing, C.J.K., R.H. and T.F.; visualization, T.F. and R.H.; supervision, C.J.K.; funding acquisition, C.J.K. All authors have read and agreed to the published version of the manuscript.

**Funding:** The investigations were partly funded by *TROTEC* GmbH, Heinsberg, Germany.

**Data Availability Statement:** Due to the amount of data, the recorded images are only available upon request. The Matlab files and the particle image count data to determine the decay rates are available as a Github repository: https://github.com/tofu85/decayRateEstimation accessed on 5 April 2023.

**Conflicts of Interest:** The investigations were partly funded by *TROTEC* GmbH, Heinsberg, Germany. The *TAC V+* mobile air cleaner was provided by *TROTEC* for the investigations. The investigations were carried out in accordance with good scientific practice. The support provided by *TROTEC* has no effect on the results presented. *TROTEC* had no role in the design of the study; in the collection, analyses, or interpretation of data; in the writing of the manuscript; or in the decision to publish the results.

## Abbreviations

The following abbreviations are used in this manuscript:

| | |
|---|---|
| ASHRAE | American Society of Heating, Refrigerating and Air-conditioning Engineers |
| DEHS | di(2-ethylhexyl) sebacate |
| HEPA | high efficiency particulate sir (filter) |
| HVAC | heating, ventilation, and air conditioning systems |
| PIV | particle image velocimetry |

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
