# Peer review of "Assessment of Mobile Air Cleaners to Reduce the Concentration of Infectious Aerosol Particles Indoors"

_atmosphere, doi:10.3390/atmos14040698_

Round 1

Reviewer 1 Report

Review: “Assessment of mobile air cleaners to reduce the concentration of

infectious aerosol particles indoors”

In their abstract, the authors summarize well the motivation of the research conducted, the content and the research questions addressed. The defined research questions are relevant in this field of research and have not been adequately addressed in previous research in the area.

Comments and questions:

Line 17: volume flow rate instead of mass flow rate

Line 21/22: ‘uncomfortable level’ is not sufficiently defined, but vague

‘air exchange rates above 6’ needs more specification or a source, it seems too high.

Line 26: COVID-19 pandemic instead of SARS-CoV-2 pandemic is commonly used

Line 54/55: Source is missing: “However, leakage flows may reduce the filtering efficiency if these masks are worn improperly.”

Line 58-60: Source is missing: “However, mouth-and-nose covers and surgical masks cannot hold back aerosol particles due to their limited filtering capability and due to leakage flows at the mask/face interface.”

Line 68/69: “An air exchange rate of 6 is a good compromise between safety and feasibility for medium sized rooms (classroom, offices, stores, restaurants).”

Comment: Here it is not obvious how the value is inferred. Since the relevance of this factor for the work is significant, a derivation to the number defined here should be discussed via common guidelines and standards, also in the context of the different environments that are not individually classified with regard to the air exchange factor.

Line 83: It should be mentioned that air cleaners have no influence on the humidity and CO2 concentration in the room and can therefore only be used in addition to natural ventilation.

Line 94: The air filter used should be introduced in chapter 2. Since things like filter, filter area, inlet and outlet geometry are of enormous importance for this paper, these critical things should be introduced briefly in chapter 2, even if referenced to another/previous paper.

Note Filter:

Filter F7 (based on obsolete standard (EN779)) must be described by valid ISO 16890 (successor of EN 779 and internationally valid).

H14 should also be introduced with EN1822, as HEPA classes in the US, for example, are defined differently (MIL-STD-282).

Results:

The consideration of thermal comfort must be introduced in more detail, as it is a central point of this work. People are differently susceptible to draughts in different parts of the body, for example, ankles, forehead, neck. In addition, the temperature conditions play a relevant role, which do not allow a general velocity value (example: summer). A more detailed introduction, definition and classification by the authors would be desirable here.

Line 190: The section on sound pressure levels is partly formulated in colloquial terms ("pressure increases significantly, making it uncomfortable to stay in the room") and does not have the scientific classification that is required for this important topic. Here, guidelines also define which values are to be aimed for, also with regard to different requirements for the rooms.

Furthermore, it is not discussed that additional air cleaners (sound sources) lead to an increase in the sound pressure level.

Which device was used to measure the sound pressure level at what distance and height?

Line 207: The unit is missing throughout for the factor k.

Line 208: Can a higher separation rate by air cleaners operated without filters result from depositions on surfaces due to the increasing convective room flow?

The results presented by the authors are very interesting and further contribute to the development of concepts for indoor spaces without preinstalled ventilation systems for pandemic times, but also for the flu season and pollen season.

Provided that the previously described points are taken into account, I consider the article to be well suited for the journal and especially the special issue.

Author Response

Please see the response in the attached pdf file.

Reviewer 2 Report

Review of Atmosphere-2319190-peer-review-v1 Assessment of mobile air cleaners to reduce the concentration of infectious aerosol particles indoors

This is a very well done paper that should be published with only minor revisions, which are suggested below. Thank for the opportunity to review this paper.

Abstract

Line 4. There might be other interventions as well, e.g. lockdowns, so perhaps the authors might want to add “and others” to the end of this sentence.

Line 5. Wouldn’t even rooms WITH HVAC systems also perhaps be amenable to mobile air cleaners?

The abstract does not contain any numerical results, such as overall filtering efficiency or the distance the air cleaner could be considered to remain effective. It might also be useful to know if the rooms were occupied (causing some resuspension of particles) or whether the air in the room was quiescent.

The keywords might include “indoor air quality”.

Line 45. It is not clear that social distancing alone will mean “people can protect themselves very effectively”

Line 86. Please correct grammar

Line 96. Not all readers will know what a H14 main filter means. Is this a HEPA filter or something else?

Line 107 Section 2 might state whether these rooms had carpeting, or were all the surfaces hard? This might affect resuspension of particulate sinks in a real world setting.

Line 138. Please correct grammar

Line 166. “Mp1-4” should be defined; are these the same thing as measurement locations?

Line 261. Should “As it is the case for” be: “As is the case”?

Line 266. Please correct grammar

Line 273. Please correct grammar

Author Response

(The authors gave the same response as above.)
